# Convergent innervations of mesencephalic trigeminal and vestibular nuclei neurons onto oculomotor and pre-oculomotor neurons— Tract tracing and triple labeling in rats

Yongmei Chen[1,2], Xinrui Gong[2,3]*, Shaimaa I. A. Ibrahim[2,4], Houcheng Liang[5], Jingdong Zhang©[4,5]*

1 Department of Central Laboratory, Hubei University of Art and Science Medical Center, Xiangyang City, Hubei, China, 2 Institute of Neuroscience, Xiangyang Central Hospital, Affiliation of Hubei University of Art and Science, Xiangyang City, Hubei, China, 3 Department of Anesthesiology, Xiangyang Central Hospital, Affiliation of Hubei University of Art and Science, Xiangyang City, Hubei, China, 4 Department of Anesthesiology, University of Cincinnati College of Medicine, Cincinnati, OH, United States of America, 5 Xi'an BRIGHT Eye Hospital, Xi'an, Shaanxi, China

* gongxrbxy@gmail.com (XG); zhang2jd@ucmail.uc.edu (JZ)

**Data Availability Statement:** http://www.xyszxyy.com/portal/departs/detail/id/14.html https://www.ncbi.nlm.nih.gov/pmc/articles/PMC3230261/ DOI:

## Abstract

In studies of vestibulo-ocular reflex (VOR), the horizontal VOR circuit is much clearer than vertical-torsional VOR. The circuit and mechanism of gravity-related vertical-torsional VOR is probably weak. "Somatosensory vestibular interaction" is a known extra source to facilitate VOR, and cervico-ocular reflex is a representative for torsional VOR compensation. Whereas, how the cervical afferents finally reach the oculomotor system is less documented. Actually, when the head tilts, which generates cervico-ocular reflex, not only the neck muscle is activated, but also the jaw muscle is stretched by gravity dragged mandible and/or tissue-muscle connection between the mandible and clavicle. We have previously identified a projection from the jaw muscle afferent mesencephalic trigeminal nucleus (Vme) neurons to oculomotor nuclei (III/IV) and their premotor neurons in interstitial nucleus of Cajal (INC)–a well-known pre-oculomotor center manipulating vertical-torsional eye movements. We hypothesized that these projections may interact with vestibulo-ocular signals during vertical-torsional VOR, because effects of gravity on jaw muscles and bones has been reported. Thus, we injected different anterograde tracers into the Vme and medial vestibular nucleus (MVN)—the subnuclear area particularly harboring excitatory vestibulo-ocular neurons, and immunostained III/IV motoneurons. Retrograde tracer was injected into the III in the same animals after dual anterograde tracers' injections. Under confocal microscope, we observed the Vme and MVN neuronal endings simultaneously terminated onto the same III/IV motoneurons and the same INC pre-oculomotor neurons. We consider that jaw muscle proprioceptive Vme neurons projecting to the III/IV and INC would sense spindle activity if the jaw muscle is stretched by gravity dragged mandible or connection between mandible and clavicle during head rolling. Therefore, the convergent innervation of the Vme and MVN neurons onto the oculomotor and pre-oculomotor nuclei would be a

10.7555/JBR.29.20150084; DOI: 10.7555/JBR.31.
20160127; DOI: 10.18240/ijo.2018.03.06; DOI: 10.
18240/ijo.2020.01.05; DOI: 10.1016/j.mehy.2020.
110210.

**Funding:** XRG 2019CFB411 Hubei Province Nature
Science Foundation https://www.nsfc.gov.cn/
english/site_1/index.html The funders had no role
in study design, data collection and analysis,
decision to publish, but encourage and support the
research article publication.

**Competing interests:** The authors have declared
that no competing interests exist.

neuroanatomic substrate for interaction of masticatory proprioception with the vestibulo-
ocular signals upon the oculomotor system during vertical-torsional VOR.

## Introduction

The vestibulo-ocular reflex (VOR) is a critical approach to stabilize visual imaging on retina in
static and dynamic states and to ensure clear vision during everyday activities. A physician suf-
fered labyrinthine loss himself recorded his own experience shortly after the lost: he had to
brace his head between two metal bars to help reading, otherwise the words would jump and
blur at the same rate as his heart-beating [1]. Studies of VOR and its underlying mechanisms
revealed that neuronal circuits executing horizontal VOR are much clearer than those con-
ducting vertical-torsional VOR, and the pathway conveying semicircular canal generated VOR
is more elucidated than that leading maculae initiated VOR [2]. The horizontal semicircular
canal or utricular macula generated horizontal VOR has been documented to be through a
typical three-order neuron reflexive arc [2–5]. While the vertical canals conducted vertical-tor-
sional VOR is probably through a three- or four-order neuron reflexive arc [6–8]. However,
the utricular and saccular maculae initiated vertical-torsional VOR is usually through a long-
latency polysynaptic pathway, and the precise location of the relay neurons are uncertain
[9,10]. Both maculae participate in the perception of gravitational and linear acceleration [2];
consequently, signals for gravity-related vertical-torsional VOR are probably weak and may
need additional inputs to drive relevant eye movement effectively [11]. Thus, the neuronal net-
work underlying the gravity associated vertical-torsional VOR is still open to explore.

A line of previous studies showed that somatosensory input such as proprioception from
limb joints, neck and/or extraocular muscles, could influence the oculomotor movements
evoked by VOR [1,12–15]. For instance, ocular counter-roll (OCR) is a static vertical-torsional
VOR characterized by torsional rotation of the eyeball in response to lateral tilt of the head or
trunk [2]. Investigators have unveiled neck proprioceptive afferents facilitating the OCR [16–
19], also termed cervico-ocular reflex, since gain of the OCR was evidently higher when the
subjects performed head tilt with trunk still or *verse visa*, compared to those performing whole
body roll [1,16,17]. And vestibular nucleus (VN) neurons have been shown to receive afferent
projections from cervical dorsal root ganglion [18], or from cervical spinal cord neurons
[19,20]. Tolu and Pugliatti had found that head rolling or stimulating the semicircular canal
afferent nerve elicited excitatory responses of masseter muscle or its motoneuron in guinea
pigs [21,22], and short latency masseter muscle activation was evoked by vestibular stimulation
in humans [23,24]. These authors focused on VN efferent to the masticatory system; while, we
inquired if there are reciprocal projections between vestibular and masticatory system, like the
reciprocal connections between the neurons innervating the neck muscle and the correspond-
ing VN neurons.

Coincidentally, we found a projection from mesencephalic trigeminal nucleus (Vme) to ocu-
lomotor and trochlear nuclei (III/IV) when we re-assessed a hypothesis on Marcus Gunn Syn-
drome [25–28]. We did not see the projection from the Vme to the VN; instead, the
projections travel directly to the III/IV, and even to their premotor neurons in the INC and
Darkschewitsch nucleus (DN). The INC is a well-established pre-oculomotor center control-
ling the vertical-torsional eye movements [29,30]. Besides, we did not detect any projection
from the Vme to abducens nucleus and their premotor neuron pool [26,27]. These facts sug-
gest that the pathway is possibly related to vertical-torsional oculomotor behavior. Considering

the ambiguous and weak connection of the maculae afferents with the vertical-torsional oculomotor system, it is possible that projection from the Vme to the oculomotor system would act as a somatosensory input to impact the vertical-torsional VOR. Significantly, simultaneous bilateral electromyographic (EMG) recording on human masseter muscles showed that firing rate was increased in one side and decreased in the other side when the subject rolls the head away from upright position, which happens alternatively if the head continues to roll from side to side [31]. The author concluded that alternative change of discharging rate correlates with the shifting of head gravity center away from its upright posture [31]. We think the phenomenon observed by Tolu and Pugliatti [22] in guinea pigs, as aforementioned, was also related to the effect of gravity on jaw muscles or bones in addition to the VN efferent innervations onto the masseter motoneurons [21].

In light of these previous studies, we propose that the gravity-related vertical-torsional eye movement signals including primary maculae afferent signals, might integrate with aforementioned Vme projections while they finally reach the oculomotor system through a convergent innervation from both the Vme afferent and excitatory vestibulo-ocular inputs. Meanwhile, it is known that the majority of the oculomotor projecting VN neurons, termed as vestibulo-ocular neurons, are located in the medial and superior VN, and the former harbors excitatory vestibulo-ocular neurons and the latter dominates inhibitory [6,7,32–34]. Thus, the last order neuron in a pathway of maculae engendered vertical-torsional VOR is supposed to be located in the medial VN (MVN), although the entire reflexive circuit is still unclear. To this end, we conducted double and triple tract tracing combined with immunofluorescent staining to explore the proposed convergent innervations of the Vme and MVN neurons onto the oculomotor and pre-oculomotor neurons; if any, which ought to be a neuroanatomical substrate for the potential interaction between afferent signals of the vestibular and masticatory system upon the oculomotor system.

## Materials and methods

Thirty-three adult Sprague-Dawley rats (300-350g, twenty-three males, ten females), from Animal Facilities of Hubei University Medical Center, were used in the experiments and divided into 3 groups as shown in Table 1. All surgical procedures and animal care were carried out complying with the "Instructions for Care of Laboratory Animals in Research" issued by the Research Office of Hubei University, which is in line with the European Union guideline for Laboratory Animal Care and Use. Experimental protocol was approved by the Institutional Research Committee.

### Combination of fluorescent Nissl stain with anterograde tract tracing

**Biotinylated Dextran Amine (BDA) injection.** Animals were administered atropine (0.15 mg/kg, *i.p.*), anesthetized with sodium pentobarbital (40 mg/kg, *i.p.*), and placed on a stereotaxic frame until no limb-withdrawal reflex was elicited by pinching the hind paw. After a craniotomy on parietal skull, a glass micropipette filled with 10% BDA (mol. Wt. 10,000, Molecular Probes, Eugene, OR, USA) in saline was advanced into the caudal Vme according to the rat brain atlas [35]. The coordinate is 0.4–1.0 mm posterior to the interaural line and 1.4–1.6 mm lateral to the midline, and 5.2–5.6 mm depth from the surface of the brain with the tip tilted 16˚ rostrally. The BDA was iontophoretically delivered with 2 Hz, 200 ms duration positive current output by an electric stimulator (Grass Pulse Stimulator S88, A-M Systems, Sequim, WA). A rhythmed lower jaw elevation, with the same rate as pulse generation, by minimum iontophoresis current is a sign of good injection to the Vme. The animals were administered analgesic (Ibuprofen syrup 20 mg/kg, orally) after surgery for 3 days.

**Table 1. Experimental groups with tracer injected and successful rates.**

| Experiment | Observed ideal tract tracing with histologic staining | Total number of rats and successful rate |
|---|---|---|
| *Single tracing group (1)* BDA@Vme | BDA labeled terminals and varicosities were observed in the ipsilateral III and INC (n = 5) | N = 8 5/8 = 60% |
| *Double tract tracing group (2.1) + Nissl* BDA@Vme + PHA-L@MVN + Nissl stain | Simultaneous contact of both BDA and PHA-L labeled terminals with Nissl stained III and INC neurons, reflecting convergent innervation of Vme and MVN neurons onto the III and INC, were observed (n = 3). | N = 7 3/7 = 40% |
| *Double tract tracing group (2.2) + ChAT* BDA@Vme + PHA-L@MVN + ChAT immunostaining | Simultaneous contact of both BDA and PHA-L labeled boutons with ChAT positive motoneurons in III, reflecting convergent innervation of Vme and MVN neurons onto the III motoneurons, were identified (n = 3). | N = 7 3/7 = 40% |
| *Triple tract tracing group (3)* BDA@Vme + PHA-L@MVN + CTB@III | BDA and PHA-L anterograde labeled terminals were seen to simultaneously contact with CTB retrograde labeled INC neurons under confocal observations, indicating Vme/MVN convergent innervating pre-oculomotor neurons (n = 3). | N = 11 3/11 = 27% |
| | Note: the result was excluded if the BDA injection spread to the locus coeruleus, or if the PHA-L injection contaminated the prepositus nucleus, or if the CTB injection core located in adjacent medial longitudinal fasciculus instead of the III. | |

BDA@Vme: BDA injected to the Vme; ChAT: Choline Acetyltransferase; CTB@III: CTB injected to the III; PHA-L delivered to the MVN; n: number of successful cases; N: total number of single or multiple injections.

**Phaseolus Vulgaris Leucoagglutinin (PHA-L) injection.** Animals were anesthetized in the same way as mentioned above, 2.5% PHA-L (VectorLabs, Burlingame, CA, USA) in 0.01M phosphate buffer saline (PBS, pH 7.4), was delivered with 2 Hz, 200 ms duration positive current output to the MVN contralateral to the BDA injection to the Vme, with a coordinate that is 0.7–1.2 mm posterior to the interaural line and the tip tilted 16˚ rostrally, 1.9–2.1 mm lateral to midline, and 6.0–6.5 mm depth from the surface of the cerebellum [35]. An evident contralateral eye up-rolling and ipsilateral eye retraction by minimum pulse current is a sign of perfect injection upon the MVN. The same dose of analgesic was given after surgery for 3 days.

**Immuno-fluorescent Nissl stain with tracers that double label the III/IV nuclei.** Nine days after BDA and PHA-L injection, the animals were euthanized with an overdose of sodium pentobarbital (about 100 mg/kg, *i.p.*) and transcardially perfused with saline followed by 2% paraformaldehyde and 0.5% glutaraldehyde in PBS for an hour. The brainstem was removed and placed in 20% sucrose in PBS overnight. Frozen sections were cut into 30 μm and collected serially from caudal to rostral in 0.01 M PBS (pH 7.2–7.4) in room temperature (RT). Sections were immune-blocked (1% normal donkey serum and 1% Triton-X 100 in 0.01M PBS) in RT for an hour and incubated with polyclonal rabbit anti-PHA-L (1:1000; Dakopatts, Denmark) overnight at 4–8˚C. Next day, the sections were transferred into a cocktail of Alexa Fluor 568 conjugated streptavidin (Molecular Probes, Eugene OR, USA), Alexa Fluor 488 linked donkey anti-rabbit and Neuro Trace Nissl dye N-21479 (Molecular Probes), and incubated for about 2 hours to visualize triple labeling. Finally, sections were mounted with Vectashield mounting medium (Vector Labs), and examined under a conventional Nikon E-600 fluorescent microscope.

## Combination of Choline Acetyltransferase (ChAT) immunostaining with anterograde tract tracing

**Anterograde tract tracer injections.** The BDA and PHA-L were delivered in the same way as described above, and the animals were treated in the same way after surgery. Then, animals were allowed to survive for nine days before euthanasia.

**Immuno-fluorescent visualization of ChAT, BDA and PHA-L triple labeling in the III/IV nuclei.** Nine days later, the animals were euthanized and perfused using the same protocol as described above. Frozen coronal sections were cut at 30 μm thickness. Sections were immune-blocked in the same way at RT for an hour, and incubated with a mixture of mouse anti-ChAT monoclonal antibody (1:200; Chemicon International, Temecula, CA, USA) and rabbit anti-PHA-L overnight at 4–8˚C. Next day, the sections were rinsed and transferred into a cocktail of Alexa Fluor 568 conjugated streptavidin, Alexa Fluor 488 bound donkey anti-rabbit and either Alexa Fluor 350 (Molecular Probes) or ByLight 405 conjugated donkey anti-mouse (Jackson Labs, West Grove, PA, USA) to visualize triple labeling of BDA, PHA-L and ChAT. The Alexa Fluor 350 is for observation by conventional Nikon E-600 fluorescent light microscope and the ByLight 405 is for visualization by Bio-Rad 1024 Laser Scan Confocal Microscope.

## Combination of BDA and PHA-L anterograde with Cholera Toxin B (CTB) retrograde tract tracing

**The retrograde tracer injection.** BDA and PHA-L were applied in the same way to the caudal Vme, and to the opposite MVN, respectively. Four days later, the same rats were re-anesthetized and 2% CTB (Sigma-Aldrich, St. Louis, MO) in saline was iontophoretically delivered in the same way as did BDA/PHA-L into the III contralateral to the BDA and ipsilateral to the PHA-L injections. The coordinates guiding the electrode to the III are 6.5–7.0 mm posterior to the bregma and 0.3–0.5 mm lateral to midline, and 6.5–7.0 mm depth from the surface of the brain [35]. An obvious eyeball bobbing, with a stronger contralateral eye bobbing by minimum iontophoresis current is a sign of correct injection onto the III. After another 5 days survival, the rats were euthanized and perfused in the same way as mentioned above. Brainstem was taken and cryo-protected using 20% sucrose as aforementioned.

**Immuno-fluorescent visualization of the triple labeling in the INC/DN.** The 30-μm coronal sections were incubated with goat anti-CTB polyclonal antibody (1:200; List Biological Labs, Campbell, CA, USA) and the rabbit anti-PHA-L overnight at 4–8˚C. Next day, the sections were incubated in the cocktail of the same conjugated 568 and 488, plus donkey anti-goat conjugated either Alexa Fluor 350 or ByLight 405 to reveal CTB labeled pre-oculomotor neurons; however, the Alexa Fluor 350 was for conventional fluorescent microscopy and the ByLight 405 was for confocal microscopy observations.

## Sexual and side difference in BDA, PHA-L and CTB injections

There was no evident qualitative difference in all anterograde and retrograde labeling no matter the tracer injection was applied to male or female animals, and no matter the injection was targeted at the left or right side of the Vme, the MVN and the III.

## Imaging acquisition with conventional and confocal microscopy and processing

Digital images of fluorescent labeling were observed and captured through Nikon E-600 fluorescent light microscope and data processed by Spot Imaging 5.1 (Sterling Heights, MI, USA). Laser scanned images were captured by Bio-Rad 1024 Laser Scan Confocal Microscope and data collected through BioRad Laser Sharp 2000 imaging program (Digital BioRad Center, Pleasanton, CA USA). Alexa Fluor 568 labeling was viewed through 561 excitation laser line with 5 nm resolution of spectra. Alexa Fluor 488 and ByLight 405 were viewed through 488 and 405 excitation laser line, also through 5 nm resolution spectra. For viewing Alexa Fluor

350 by conventional Nikon E-600, 350/30 Ex Filter was applied. While, N-21479 Nissl stain was imaged through Ex-435/Em-455 filter under Nikon E-600. Confocal images were captured with 20X and 60X objective lens at iris of 2.0–2.5 in box size of 1024×1024. All Z-scan was set up at 2 μm layer of laser scan step and processed by Laser Sharp 2000. Files in Laser Sharp formula were converted to Photoshop 7.0.1 (Adobe, CA, USA) through Bio-Rad Plug-In software and saved as "tiff" files at 1024×1024 pixels.

## Results

### Nissl or ChAT, BDA and PHA-L triple labeling in the III/IV nuclei

**1. Projections from the Vme and MVN to the III/IV.**  The BDA labeled Vme projecting fibers and terminals distributed longitudinally along the cell column of the III/IV and the distribution pattern in a representative case is shown in Fig 1. Generally, anterograde BDA labeling scattered relative evenly in coronal planes at the rostral part of the III (Figs 1A and 3A). At the rostral levels above the middle point of the III, the labeled fibers and terminals were usually spread in ventral or ventromedial parts of the nucleus (Fig 1B). Caudally, at the middle coronal planes of the III, the BDA positive fibers and boutons were mostly located in ventral divisions of the nucleus (Figs 1C and 3B). More caudally at the levels above the IV, the BDA labeling mostly distributed in ventral or ventrolateral parts of the nucleus (Fig 1D). The labeled fibers and boutons in the IV appear to be evenly distributed in the coronal planes (Figs 1E and 3C). The distributive pattern is similar if tracer injection is restricted to the caudal Vme (Figs 1F, 2A and 5D).

Projections from the MVN to the III/IV, including medial longitudinal fasciculus (mlf), was representatively displayed (Fig 2C–2E) in a case of excellent injection that covers both parvocellular and magnocellular division but is restricted in the rostral MVN (Fig 2B). In summary, labeled fibers and terminals were rather evenly scattered in the contralateral rostral III (Fig 2C). There were sparse PHA-L labeled terminals in the ipsilateral III except for a few labeled fibers in the surrounding area (Fig 2C). At the middle levels, heavily labeled fibers and terminals were situated in the contralateral III, and bundle like structures were broadly distributed ventral-laterally in or around the mlf (Fig 2D). Ipsilaterally, the labeling was much less than that in contralateral III with few labeled fibers in the mlf (Fig 2D). Fibers crossing midline (arrowheads in Fig 2D) to the opposite III were sometimes seen in these coronal sections. Caudally at the IV levels, more densely labeled terminals were distributed in the contralateral nucleus than in the ipsilateral side, and bundles were scattered in the mlf of both sides with more of them in the contralateral side (Fig 2E).

**2. Convergent termination of Vme and MVN projections onto the III/IV neurons.** Overlapping of BDA positive fibers and boutons from excellent Vme injection (Fig 2A) with PHA-L labeled fibers and terminals from successful MVN injection (Fig 2B) were frequently observed in both III and IV (Figs 2F, 2G and 3A–3C). Then, the III/IV neurons were visualized by Nissl staining, and BDA (arrowheads) and PHA-L (arrows) positive boutons were observed to contact the same Nissl stained neuronal somata simultaneously in the III/IV ipsilateral to BDA and contralateral to PHA-L injections (Fig 2F and 2G). The image in Fig 2F was taken from middle level of the III and it was cropped from the ventral part of the nucleus.

**3. Convergent innervation of the Vme and MVN neurons onto the III/IV motoneurons.**  In order to verify whether the III/IV motoneurons receive convergent innervations from the Vme and MVN projections, we performed ChAT immunostaining combined with BDA and PHA-L tract tracing (Fig 3A–3C), and observed via a laser scan confocal microscope (Fig 3D–3F). The confocal images clearly showed that Vme neuronal terminals (arrowheads)

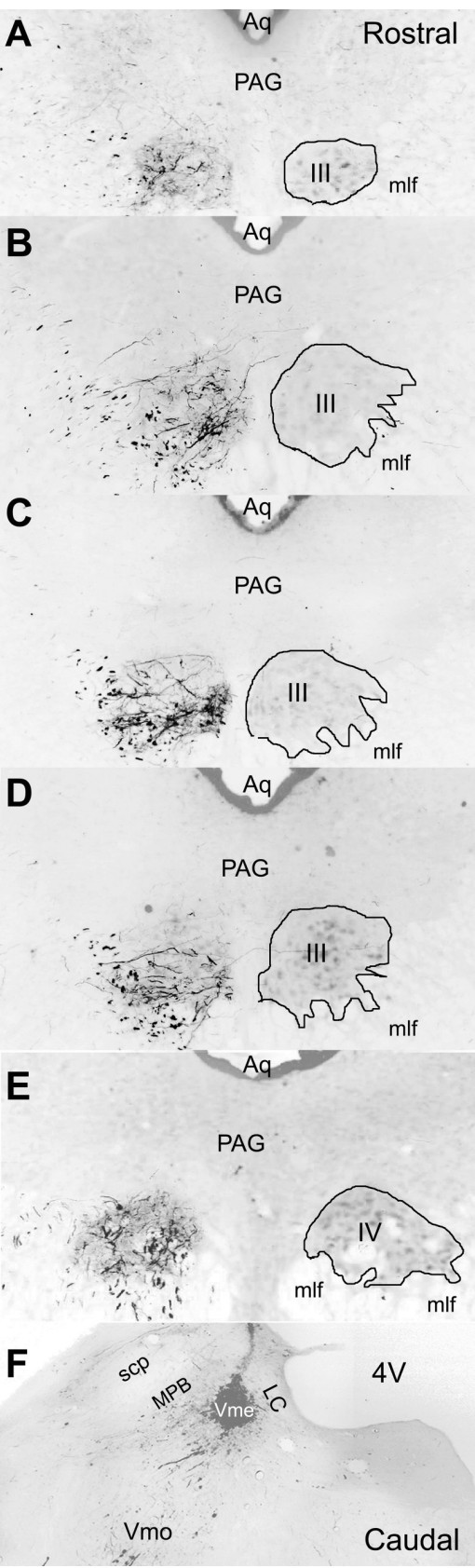

**Fig 1. Distribution of Vme projecting fibers and terminals in the III/IV.** The BDA labeled endings distribute longitudinally along the cell column of the III/IV (**A** ~**E**) following tracer injection into the caudal Vme (**F**). **A,** BDA labeled endings are scattered at rostral top of the III without consistent distributive preference. **B,** at the levels between top and middle planes, the labeled endings are preferentially distributed in ventral and ventromedial part of the III in the coronal sections. **C,** caudally around middle levels, the positive labeling are mostly at ventral and ventrolateral division of the nucleus. **D,** similarly, at more caudal levels above the IV, the BDA labeling preferentially distributed in ventral or ventrolateral nucleus in the coronal planes. **E,** finally in the IV, labeled endings seem to be evenly distributed. **F,** it is clear that the BDA injection site is restricted in the caudal Vme. 4V, the fourth ventricle; Aq, aqueduct; LC, locus ceoruleus; mlf, medial longitudinal fasciculus; MPB, medial parabrachial nucleus; PAG, periaqueductal gray; scp, superior cerebellar peduncle; Vmo, trigeminal motor nucleus.

and MVN projecting axon boutons (arrows) simultaneously contact the same ChAT positive motoneurons in the III (Fig 3D and 3E) and IV (Fig 3F).

## BDA and PHA-L anterograde tracing combined with CTB retrograde labeling in the INC/DN

**1. Overlapping of BDA positive fiber-boutons with PHA-L labeled ones in the INC/ DN.** Following injection of the tracers into the MVN (Fig 2B), the majority of the PHA-L anterograde labeled fibers and boutons were distributed in the contralateral INC/DN with a limited amount of positive labeling in the ipsilateral nuclei. The labeling was mostly situated in the INC territory based on the rat brain atlas [35], and a small portion of the labeling was spread medially in the DN area and outside of the INC ventrolaterally (Fig 4A–4C, middle column). This distribution pattern of vestibular INC/DN projection is similar to that previously observed in rabbits and cats [6,7,36]. The Vme projections to the INC/DN, namely the BDA labeled fibers and terminals, were restricted in the INC/DN ipsilateral to the Vme injection. Similarly, the BDA labeled fibers and boutons were densely distributed in the INC and less dense in the DN area (Fig 4A–4C, right column). Consequently, the Vme neuronal fibers and boutons closely overlapped with those projections from the contralateral MVN (Fig 4A–4C, left column), inferring the convergent innervations of both Vme and MVN neurons onto the pre-oculomotor neurons in the INC/DN.

**2. Convergent innervation onto the pre-oculomotor neurons by Vme and MVN projections.** Following visualization of the CTB by blue anti-CTB immunofluorescent staining, the injection site was confirmed to be confined in the III (Fig 5A) and the retrograde labeled pre-oculomotor neurons were only seen in the contralateral INC/DN (Fig 5B and 5C), which is consistent with our previous works [26,27,37]. Of note, the same animal had received BDA and PHA-L injections onto the Vme (Fig 5D) and MVN (E) before the CTB injection (A). Therefore, triple immunofluorescent staining revealed that BDA and PHA-L positive varicosities were precisely overlapped with CTB labeled somata in the INC/DN region (Fig 5F). Then, simultaneous termination of the Vme and MVN neuronal endings (arrowheads and arrows) onto the dendrite (Fig 5G) or the soma (Fig 5G and 5H) of the same pre-oculomotor neurons in the INC was imaged under a confocal microscope.

## Discussion

The present work unveiled a convergent innervation from both the Vme and MVN neurons onto the motoneurons in the III/IV and their premotor neurons in the INC/DN. Considering the Vme projections to the III are largely distributed in ventral parts of the nuclear column, we first analyzed which group of ocular motoneurons are most likely receiving the projections from the Vme neurons. It was documented in rats that the motoneurons in the ventral part of the rostral III innervate the ipsilateral inferior rectus, and of the middle and caudal nucleus

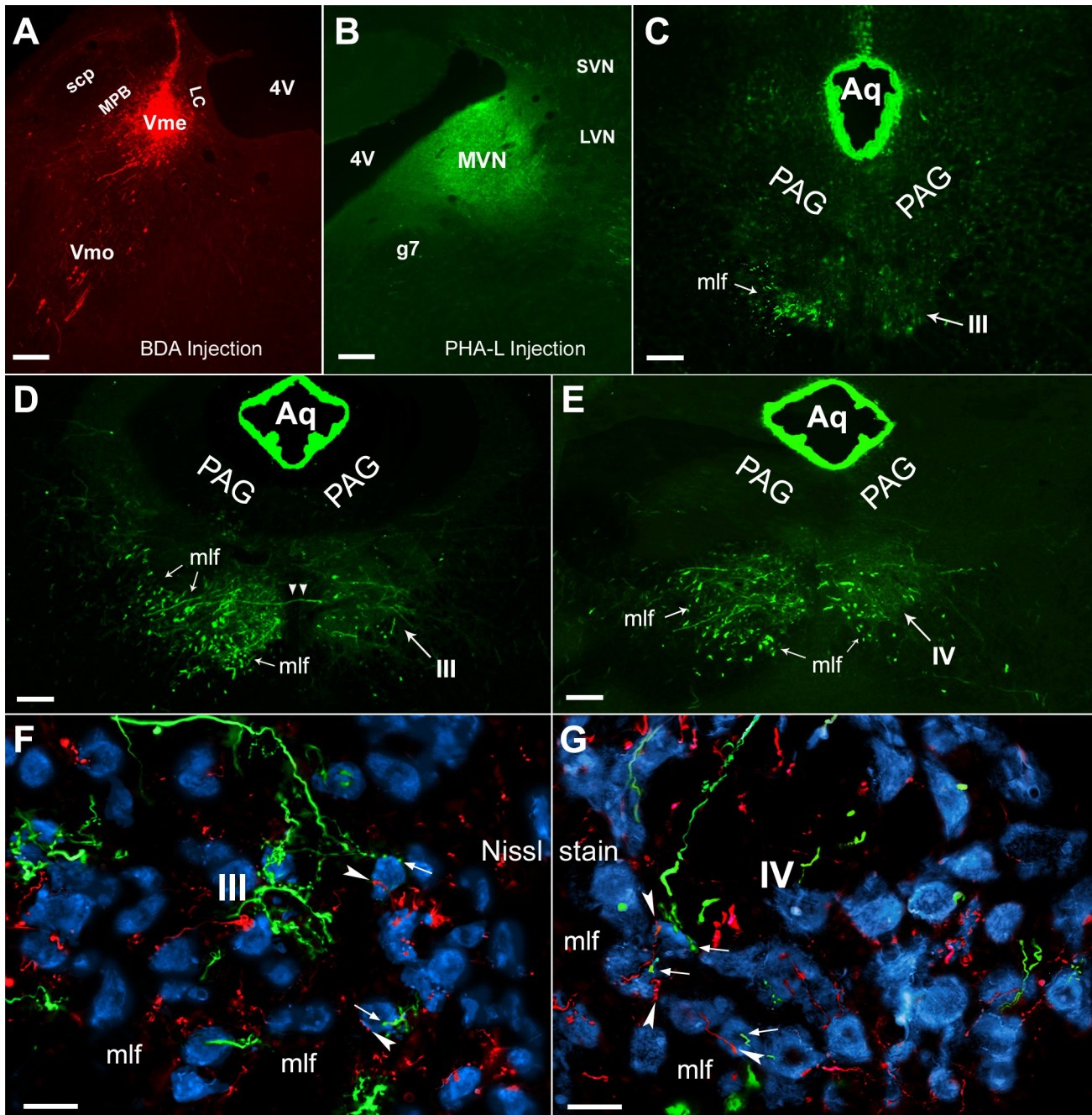

**Fig 2. Distribution of MVN originated—PHA-L labeled fibers and terminals in the III/IV and triple labeling of BDA and PHA-L with Nissl staining. A**, BDA injection site at the caudal Vme. **B**, PHA-L injection site in the rostral MVN. **C,** in rostral part of the III, PHA-L positive fibers and terminals were seen sparsely in both sides of the III, with densely labeled bundles in the contralateral mlf. **D,** at the middle levels of the III column, heavily labeled fibers and terminals were situated in the III contralateral to the MVN injection, and bundle like structures were broadly and densely distributed in the mlf ventral-lateral to the III. Lightly labeled fibers were observed in ipsilateral III with few bundle-like structures in the mlf area. Some fibers crossing midline (arrowheads) to the opposite III were occasionally seen in these coronal sections. **E,** caudally in the IV levels, bundle like structures were scattered in the mlf of both sides with more bundles surrounding the contralateral IV. In the IV, more densely labeled terminals were located in the contralateral nucleus than in the ipsilateral side. **F** and **G**, convergent projections from both the Vme (red) and MVN neurons (green) onto the Nissl stained (blue) neurons (arrowheads and arrows) were observed in the III/IV. g7, genu of facial nerve; MVN, LVN and SVN: medial, lateral and superior vestibular nuclei. Scale bars in **A ~ E** are 200 μm, in **F** and **G** are 50 μm.

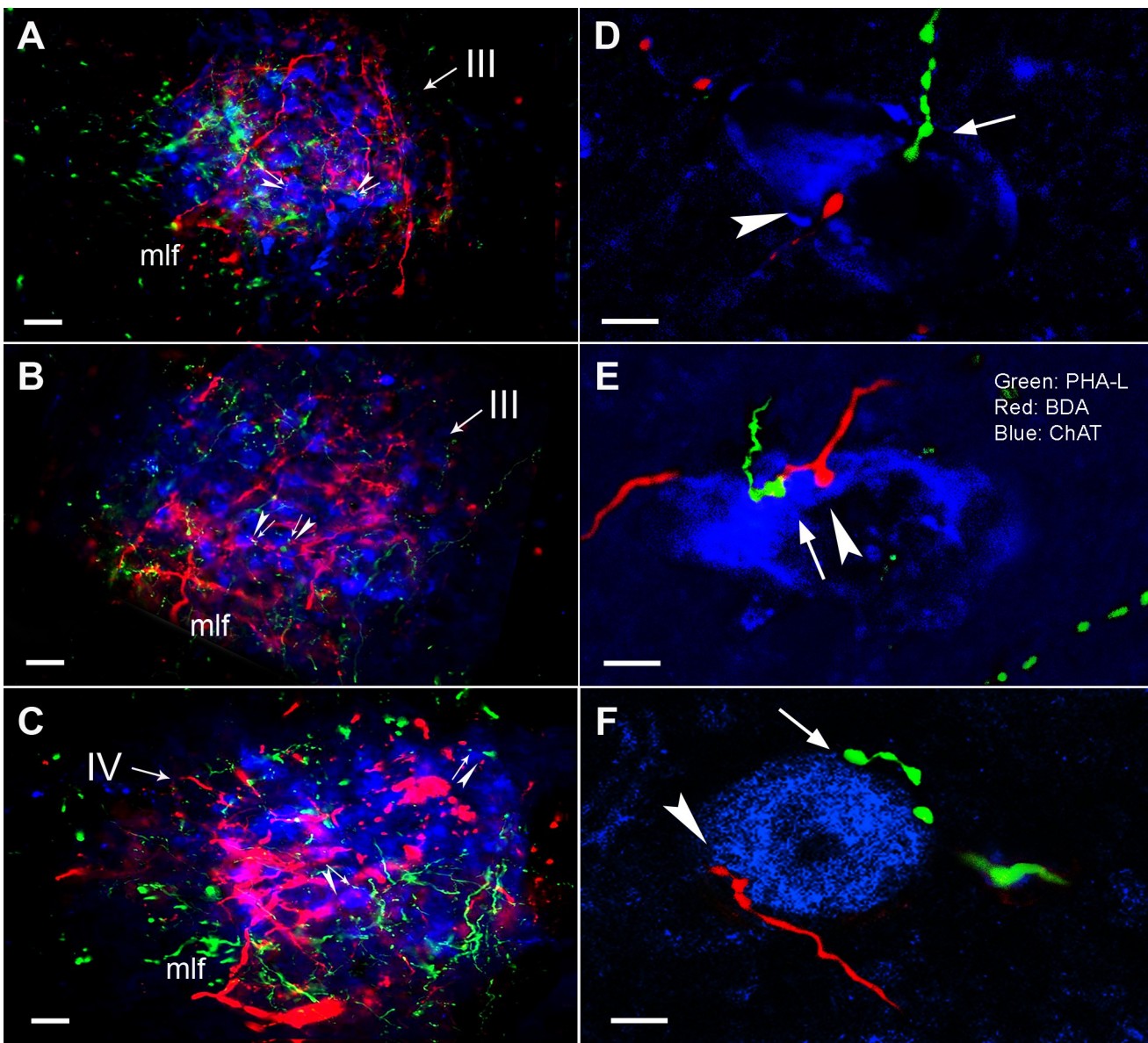

**Fig 3. Convergent innervations from Vme and MVN neurons onto ChAT immune-stained III/IV motoneurons. A** and **B**, overlapping of BDA positive (red) Vme projecting endings and PHA-L labeled (green) MVN projecting terminals with ChAT immuno-stained (blue) motoneurons (arrowheads and arrows) in the III. **C**, the same overlapped triple labeling in the IV. **D** and **E**, confocal imaging showing simultaneously immediate contact of both Vme and MVN projecting terminals (arrowheads and arrows) upon the ChAT immuno-stained motoneurons in the III. Further, most of the convergent innervations from the Vme and MVN projections onto the single motoneurons were visualized in ventral parts of the III. **F**, the same convergent terminations from the Vme and MVN projections onto the IV motoneurons (arrowhead and arrow). Scare bars in **A ~ C** are 50 μm, in **D ~ F** are 10 μm.

innervate the contralateral superior rectus plus bilateral levator palpebrate [38–40]. The IV was known to innervate the superior oblique alone [2]. Hence, these ventral-positioned ocular motoneurons, without excluding the possibility of the other ocular motoneurons, may receive projections from the Vme neurons. Secondly, the INC, DN and nucleus of posterior commissure have been designated as accessory oculomotor nuclei [36]. Upon delivery of retrograde tracer into the III, the INC/DN was always labeled as a single group of neurons [26,27,37]. Further, the INC has been demonstrated to be a pre-oculomotor center mainly controlling vertical-torsional gaze during static and dynamic head and/or body movements [29,30,36]. The

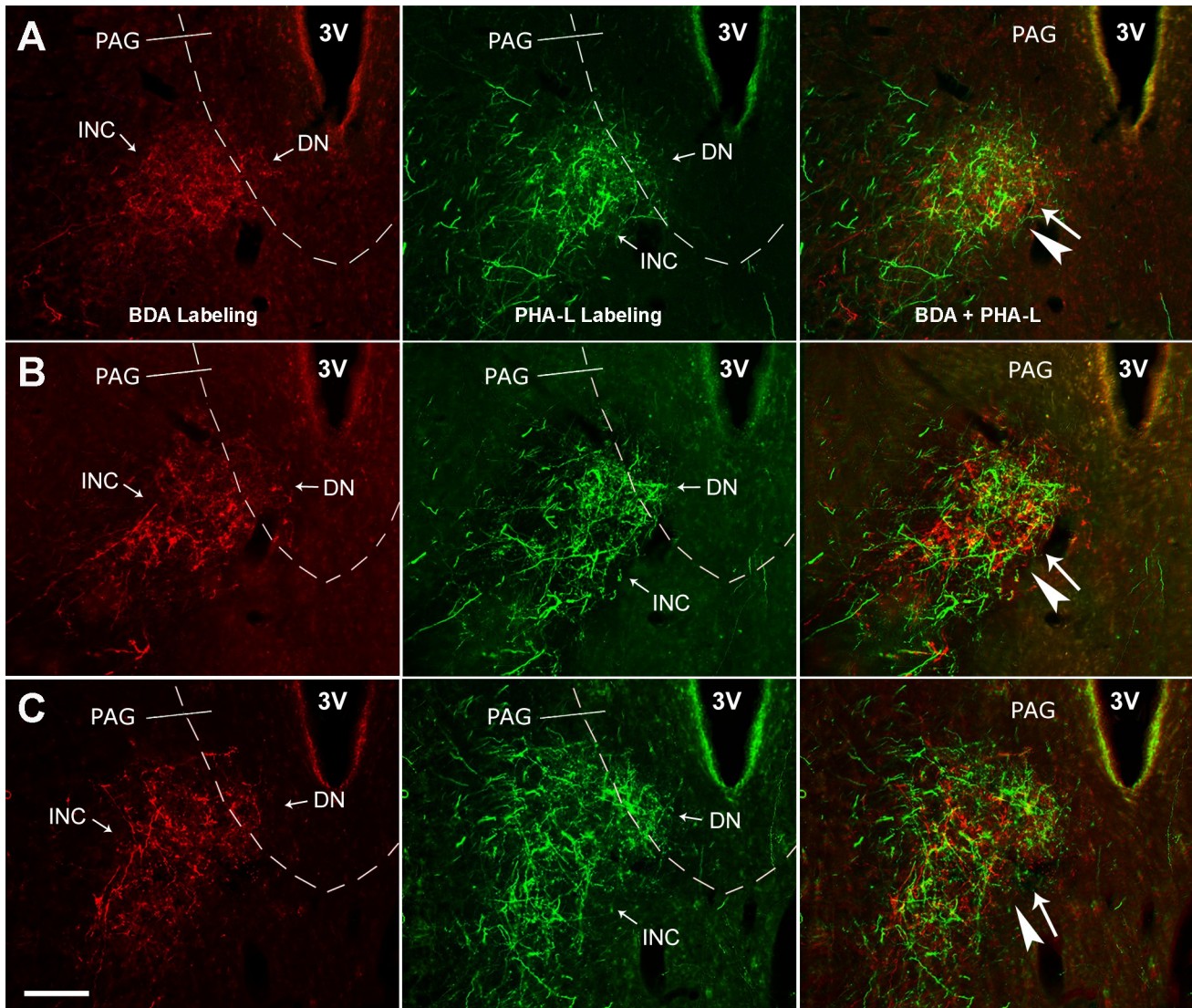

**Fig 4. Overlapping of the Vme and MVN neuronal terminals in the INC/DN. A—C**, following BDA and PHA-L injections into the Vme and the contralateral MVN (Fig 2**A** and 2**B**), the anterograde labeled fibers and boutons by both tracers were observed closely overlapped (arrowheads and arrows) in all rostral (**A**), middle (**B**) and caudal (**C**) levels of the INC/DN. 3V, the third ventricle. Scare bar is 200 μm, applying for all microimages in the figure.

distribution pattern of the Vme projections, and the convergent innervation unveiled in the current work, suggest this pathway be associated with the vertical and torsional eye movements and probably be involved in facilitation of a relatively weak vertical-torsional VOR. A proposal that these convergent innervations may assist gravity-related OCR is based on the involvement of primary jaw muscle proprioceptive afferent Vme neurons in the pathway (Fig 6) [25,26], which may represent a new source of proprioception interaction with the VOR. However, the periodontal afferent Vme neurons could not be excluded since they are predominantly located in caudal part of the Vme [41], the area we injected the tracer. While, we considered jaw muscle afferent Vme neurons herein only because we had evidences that those neurons are involved in the related pathway [25–27].

A function of muscle spindle is to generate impulses when the muscle fibers are elongated by a gravity-dragged structure, or when the gravity-dragged structure needs to be stabilized

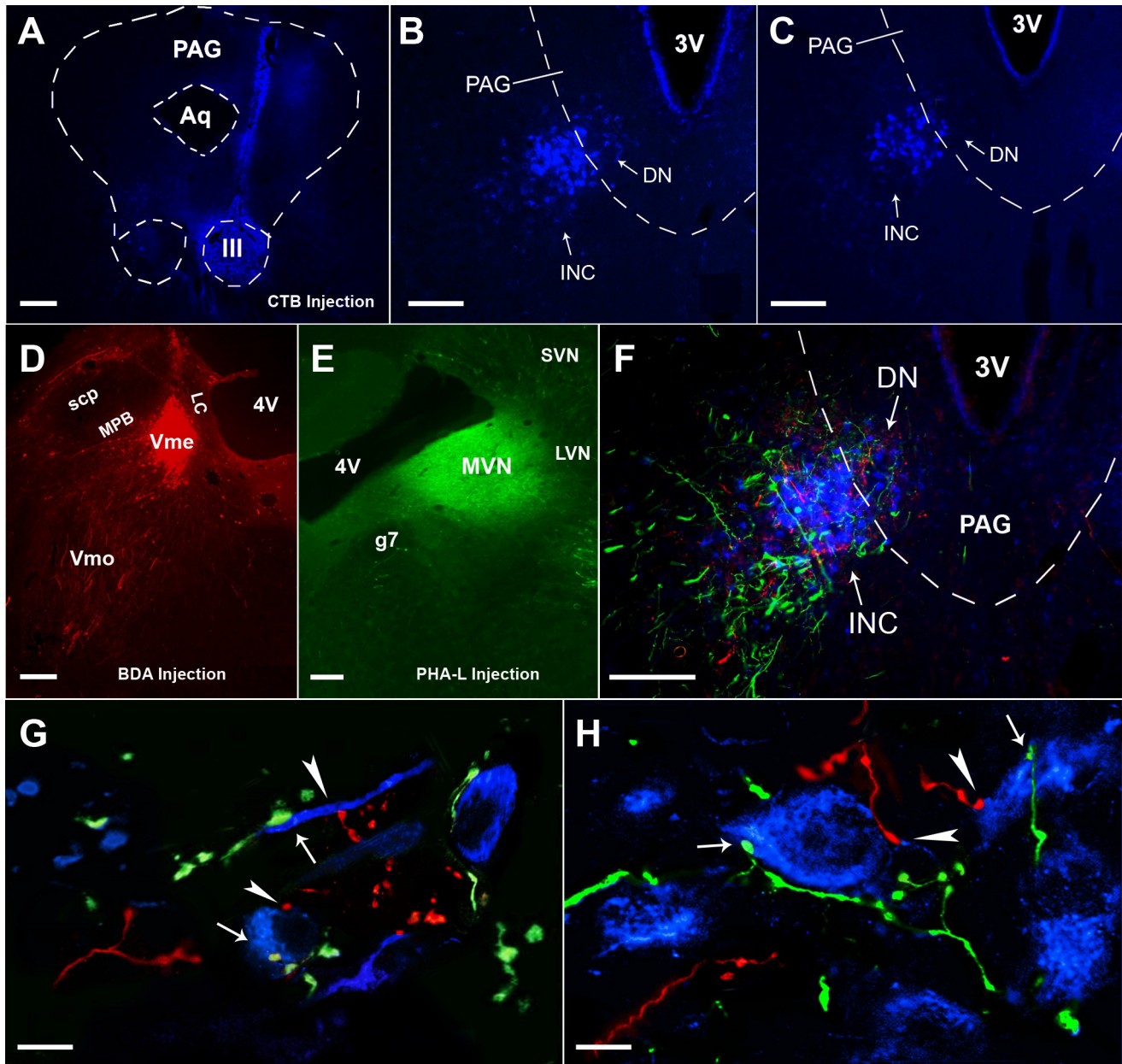

**Fig 5. Convergent innervations onto the INC/DN pre-oculomotor neurons by Vme and MVN projections. A**, CTB injection site in the III. **B** and **C**, retrogradely labeled (blue) premotor neurons of the III in the contralateral INC/DN. **D**, BDA injection site in the Vme of the same rat. **E**, PHA-L delivery site in the MVN contralateral to BDA injection. **F**, overlapping of anterograde tracer labeled axons and endings from the Vme and MVN neurons with retrograde tracer labeled pre-oculomotor neurons in the INC/DN area. **G** and **H**, confocal imaging revealing some simultaneously immediate contact of the Vme and MVN neuronal terminals (arrowheads and arrows) onto the CTB labeled dendrite or somata of the pre-oculomotor neurons in the INC area. Scare bars are 200 μm in **A**—**F**, and 10 μm in **G** and **H**.

through isometric contraction of relevant muscles [42]. In case of the jaw structures, there is a debate on whether the mandible is fixed by the jaw muscle actively working against gravity or by viscoelastic property of the tissues at its rest position, *i.e.* head upright along the yaw axis [43]. Nevertheless, it is evident that the jaw muscles are activated when the head starts to move like tilting or pitching, or like oscillating vertically or along the Frankford plane [22,31,43,44]; for example, when the head is shaking during locomotion [43,44]. As mentioned in the

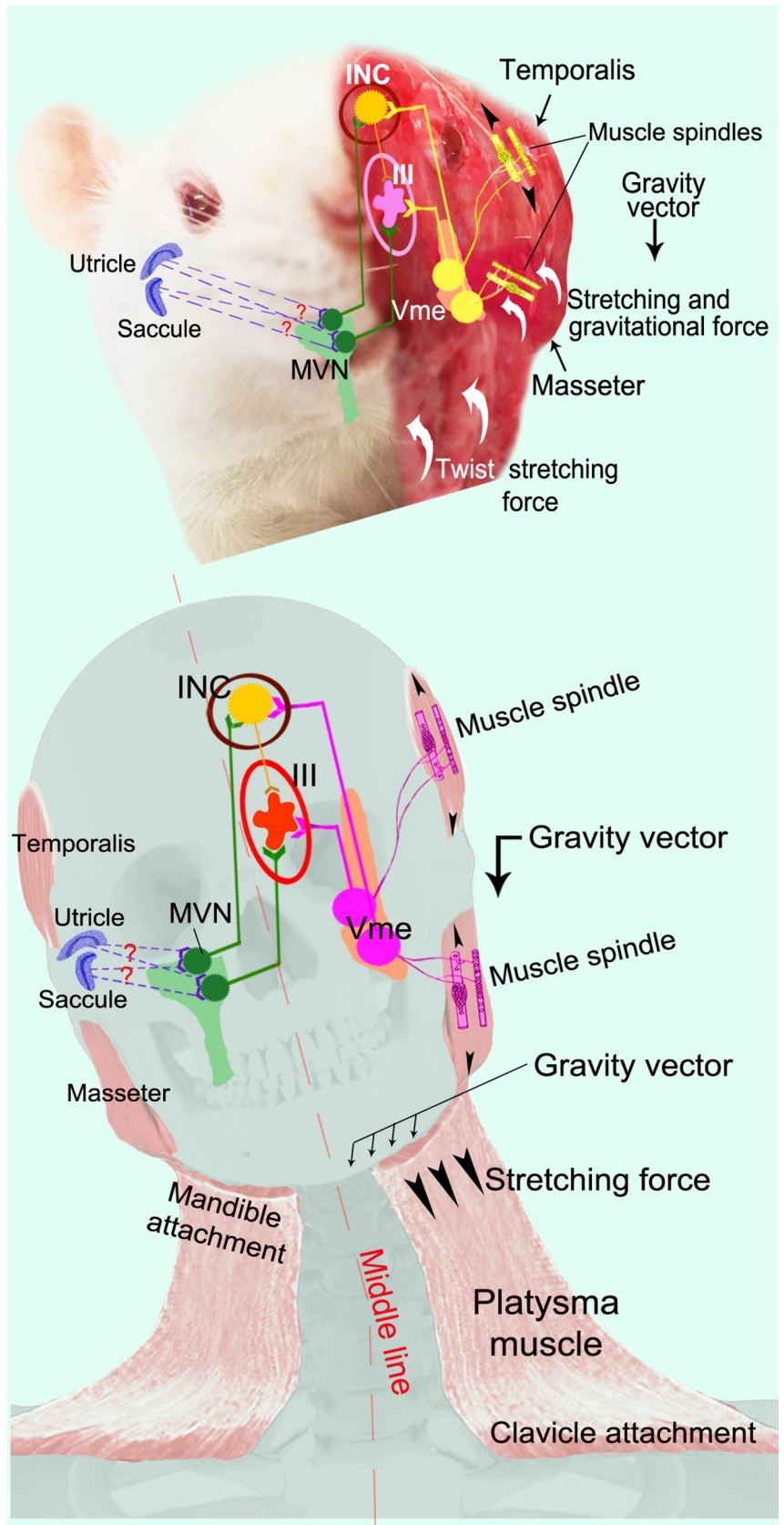

**Fig 6. Schematic diagram for potential function of convergent innervation of the Vme and MVN neurons onto the oculomotor system. A**, arrowheads and curved arrows indicate the direction of stretching force once the head rolling. The masseter muscle fibers in guinea pig or rat are extending predominantly along the roll axis and less impacted by gravity, comparing to that in humans. Hence, the masseter would be twist stretching while the head is rolling to opposite, so would be the neck muscles in these animals. The temporal muscle in guinea pig or rat is a principal muscle impacted by gravity. **B**, similarly, arrowheads indicate the direction of the potential executive force to pull the muscle spindles. In humans, the neck muscles would be stretched vertically, instead of twist along the roll axis, while the head is rolling. Moreover, there is a platysma muscle in human, but we are not sure whether guinea pig and rat have this muscle or not. This muscle directly pulls down the mandible (rowed arrowheads) in addition to gravity, when the head is tilted away. In case of this muscle stretching, we think the viscoelastic property of all the muscle, connective tissues and skin may be involved to form a combined force. Dotted lines and question marks between the otolith organs and MVN represent the proposed neural pathway between them since neural connection between maculae and vestibulo-ocular neuron is still an enigma.

introduction, when Tolu and Pugliatti [22] recorded masseter motoneuron discharging in guinea pig, they observed the firing rate was alternatively changed during animal's head rolling [22]. In humans, Ishii directly recorded EMG from bilateral masseter muscles and disclosed that the frequency of EMG alternatively increased on one side and decreased in the opposite side when the head kept rolling [31]. The author concluded that the alternative firing rate change correlates with the shifting of head gravity center away from its upright posture [31]. We think this conclusion is also appropriate for explaining the phenomenon observed in guinea pig by Tolu and Pugliatti [22]; meanwhile, we did not exclude the effect of vestibular afferents on the masseter motor units.

Consequently, the temporomandibular joint on one side will be raised up in opposition of gravitational vector when the head is tilting and the relevant jaw muscle would be stretched by gravity and inertia or viscoelastic force of the tissues and bones. The masseter of guinea pig or rat would be stretched mostly by twisting force and slightly by gravity when animal's head rolling, due to oblique extension (along roll axis) of that muscle fibers (Fig 6A). In this circumstance, the spindle intrafusal fiber would be activated and the connected jaw muscle afferent Vme neurons would be discharged. Sequentially, the trigeminal motoneurons would be excited, as recorded by Tolu and Pugliatti [21], via the innervation of the Vme neuronal central processes [45]. In the meantime, it is possible that these motoneurons are also activated through the VN neuronal innervations as Tolu and Pugliatti stated [21,22]. Moreover, evidence for Vme neurons participating in convergent innervations onto the oculomotor system are the jaw muscle afferent neurons had been described in our previous papers [26,37], as the intracellularly labeled Vme neurons that send axons to the III responded to the stimulation of the masseter nerve. Hence, we can imagine that the effect of gravity, inertia or viscoelastic force on the jaw tissue or bone during head rolling, shaking or pitching would more or less distort the masseter or temporal muscles, which in-turn evokes the jaw muscle spindle afferent Vme neurons that project to the III/IV or INC/DN (Fig 6B). Therefore, when the subjects perform head tilt with trunk still, or *vice versa*, not only the neck but also the jaw muscles, such as masseter and temporal muscles, are stretched (Fig 6) [16,17]. However, how can we explain that the gain of the OCR is smaller when the head and trunk are tilted together [16,17], considering that the effect of gravity on different sides of the jaw muscle and bone is also different in this situation? In fact, there is a broad and thin platysma muscle connecting the mandible and clavicle bones in human (Fig 6B) [46]; this muscle would drag the mandible down once the unilateral temporomandibular joint is raised up during the head rolling with the trunk fixed, or *vice versa* (Fig 6B). In the guinea pig or rat, one side of neck muscles would be twist stretching when the head is rolling to the opposite side with the trunk still (Fig 6A).

On the other hand, it is likely that this pathway is a backup or a subsidiary circuit without valid function in normal conditions, which might be more useful if the labyrinthine afferent

has malfunctioned or is damaged, such as in cases of traumatic labyrinth injury or labyr-inthectomy. After inner ear lesions or labyrinthectomy, the abnormal ocular motor behaviors or the VOR could be recovered to a certain extent, which is termed as vestibular compensation [47]. Generally, the static deficit of the VOR would be compensated or become almost normal in a shorter or longer time in different species; whereas, the dynamic VOR disorder could hardly be recovered [47–50]. Further, anomaly of OCR and head tilting at rest, especially the ocular skew deviation, are common signs of maculae injury [48,50]. These disorders could be usually recovered in a few days or within a couple of weeks after the labyrinth lesion or removal, although recovery in a fairly longer time is also encountered [47–50]. The mechanism of the compensation is still unclear but the compensation is considered to be attributed to the plasticity in the central nervous system because peripheral vestibular organs could not be regenerated [47], especially in cases of labyrinthectomy. One of the potential compensatory mechanisms is through a cervico-ocular reflex [51] as aforementioned. A couple of previous studies on monkeys and humans demonstrated that cervico-ocular reflex contributed 2–3% of ocular stabilization during rapid head movements in normal subjects and the contribution increased to 30–80% in those suffered from labyrinthectomies [1,52].

The neuronal circuit backing-up the cervico-ocular reflex has been reported, but most of the neuroanatomical data showed reciprocal connections between the VN and cervical spinal cord or dorsal root ganglion neurons [18–20]; while, how those cervical afferents connect with oculomotor neurons is still not clear. The convergent innervation showed in the current work (Fig 6) might be a substitute pathway that could also transfer the head tilting information, but convey signals directly to the oculomotor and pre-oculomotor neurons. More interestingly, Long-Evans rats subjected to a 2G centrifugal force rolling rotation exhibited Fos protein expression in the INC/DN area at 2- or 3- weeks post-labyrinthectomy [53]. We have observed that repeated down stretching of the lower jaw could elicit Fos expression in the INC/DN area in more than half of the normal Sprague-Dawley rats that are used in the experiment [37]. It is known that following functional related stimulation, *c-Fos* oncogene and Fos proteins are usu-ally expressed along the functional pathway [54]; for instance, taste related stimulation could induce Fos expression along the visceral sensation central pathway [55,56]. Since both head rotation and lower jaw movements may excite muscle spindles of both jaw and neck muscles, it is possible that both proprioceptive afferents are involved in this functional pathway. In light of these previous studies and our own observations, we think that the convergent innervation of the Vme and MVN neurons onto the III/IV and INC/DN may be involved in the OCR recovery after the labyrinth lesion or removal.

In summary, we assume the convergent innervation of the Vme and MVN neurons onto the III/IV and INC/DN may interact or facilitate the maculae afferent to the oculomotor sys-tem during vertical-torsional VOR, due to the following findings: 1, the pathway relaying the primary maculae afferent signals to the vertical-torsional oculomotor system is vague and probably weak [9–11]; 2, the mandible that is dragged by gravity, inertia or its connection to the clavicle through platysma muscle would stretch the jaw closing muscles during head rolling (Fig 6) and 3, the jaw muscle spindle would be activated once the muscle is stretched and the primary Vme neurons would transmit the signals to the III/IV and INC/DN (Fig 6). Thus, not only the vestibular system exerts an effect on the masticatory motoneurons and muscles [21,22], but also the masticatory muscle afferents would interact with the vestibulo-ocular sig-nals during vertical-torsional VOR. Further, this convergent innervation onto the vertical-tor-sional oculomotor system add a new explanation for cervico-ocular reflex (Fig 6) since anatomical data is sparse for how the reflexive impulse reach the oculomotor system; or this convergent innervation represents a new somatosensory vestibular interaction network in addition to previously reported proprioception interfering of VOR [12–15]. Moreover, it is

rational to consider that this convergent innervation may be involved in the mechanism of vestibular compensation following labyrinth injury or loss, and may especially facilitate the recovery of vertical-torsional VOR, such as the OCR. The demonstration of this convergent innervation in the current work probably added a new topic in future studies on the pathway of gravity-related vertical-torsional VOR, and the mechanism for its functional compensation when the vestibular organs, especially the maculae, are damaged.

## Acknowledgments

We warmly thank Dr. Cinder Cohen, from the Department of Anesthesiology, University of Cincinnati College of Medicine, for her critical reading and editing of the manuscript.

## Author Contributions

**Conceptualization:** Houcheng Liang, Jingdong Zhang.

**Funding acquisition:** Xinrui Gong.

**Investigation:** Yongmei Chen.

**Writing – review & editing:** Shaimaa I. A. Ibrahim, Jingdong Zhang.

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
