## [Decision Letter · Decision Letter 0]

22 Jul 2022

PONE-D-21-39088Convergent innervations of mesencephalic trigeminal and vestibular nuclei neurons onto oculomotor and pre-oculomotor neurons --- tract tracing and triple labeling in ratsPLOS ONE

Dear Dr. Zhang,

Thank you for submitting your manuscript to PLOS ONE. After careful consideration, we feel that it has merit but does not fully meet PLOS ONE’s publication criteria as it currently stands. Therefore, we invite you to submit a revised version of the manuscript that addresses the points raised during the review process.

Please note that we have only been able to secure a single reviewer to assess your manuscript. We are issuing a decision on your manuscript at this point to prevent further delays in the evaluation of your manuscript. Please be aware that the editor who handles your revised manuscript might find it necessary to invite additional reviewers to assess this work once the revised manuscript is submitted. However, we will aim to proceed on the basis of this single review if possible. 

We look forward to receiving your revised manuscript.

Kind regards,

Vanessa Carels

Staff Editor

PLOS ONE

Journal Requirements:

2. Thank you for stating the following financial disclosure: "XRG

2019CFB411

Hubei Province Nature Science Foundation

https://www.nsfc.gov.cn/english/site_1/index.html

No"

Please state what role the funders took in the study.  If the funders had no role, please state: "The funders had no role in study design, data collection and analysis, decision to publish, or preparation of the manuscript.

Reviewers' comments:

Reviewer's Responses to Questions

**Comments to the Author**

1. Is the manuscript technically sound, and do the data support the conclusions?

Reviewer #1: Yes

2. Has the statistical analysis been performed appropriately and rigorously? 

Reviewer #1: N/A

3. Have the authors made all data underlying the findings in their manuscript fully available?

Reviewer #1: Yes

4. Is the manuscript presented in an intelligible fashion and written in standard English?

Reviewer #1: Yes

5. Review Comments to the Author

Reviewer #1: The study by Chen et al., provides a morphological characterization showing the convergent projection from the mesencephalic trigeminal nucleus (Vme) and medial vestibular nucleus (MVN) onto motorneurons of the III and IV nuclei, as well as onto pre-motorneurons of the interstitial nucleus of Cajal (INC) and Darkschewitsch nucleus (DN).

The study in general is very well done. Combination of anterograde, retrograde and immnunostaining provides convincing experimental evidence supporting the conclusion that motorneurons and premotorneurons receive convergent innervation from the Vm and the MVN. This conclusion is supported by confocal microscopy showing presynaptic buttons from Vme and MVN neurons contacting the same cell body or dendrite, rising interesting perspectives on how these postsynaptic neurons integrate these inputs to control eye movements. Despite of that, some concerns arise.

Major concerns:

Anterograde tracer (BDA) injected in the Vme yields a strong staining of the ipsilateral III and IV motor neuclei, as well as of the INC and DN. However, these injections yield only very weak staining of the trigeminal motor nucleus (Vmot), as shown in Figures 1F, 2A and 5D. This is surprising as Vme afferents (particularly spindle afferents) heavily project to the ipsilateral Vmot. This monosynaptic projection is very well documented in rodents, both electrophysiologically (Grimwood et al. 1992) and morphologically (Dessem et al. 1997; Luo and Li 1991; Luo et al. 1991; Stanek et al. 2014). Authors should provide an interpretation of this unexpected results of their experiments.

Results are discussed assuming that stained Vme neurons are only spindle afferents. However, the Vme also contain a large population of afferents innervating the periodontal ligament (Shigenaga et al. 1988). Moreover, the caudal portion of the Vme, where BDA injections were performed in this study by Chen and co-workers, contain a large proportion of these periodontal afferents. In that sense, most probably some of the anterogradely stained Vme afferents belong to this class. This possibility should be taken into consideration in the Discussion section.

Minor concerns:

For consistency capitalize letters identifying panels in Figure 1.

Line 66: citation “2” should be between square brackets.

Lines 72-73: citations “16-19” should be between square brackets.

Line 120: citation “35” should be between square brackets.

Line 222: citation “35” should be between square brackets.

Line 249: should read “nucleus of posterior commissure”

Line 293: citation “46” should be between square brackets.

References

Dessem D, Donga R, Luo P. Primary- and secondary-like jaw-muscle spindle afferents have characteristics topographic distributions. J Neurophysiol 77: 2925–2944, 1997.

Grimwood PD, Appenteng K, Curtis JC. Monosynaptic EPSPs elicited by single interneurones and spindle afferents in trigeminal motoneurones of anaesthetized rats. J Physiol 455: 641–662, 1992.

Luo P, Li J. Monosynaptic connections between neurons of trigeminal mesencephalic nucleus and jaw-closing motoneurons in the rat: an intracellular horseradish peroxidase labelling study. Brain Res 559: 267–275, 1991.

Luo PF, Wang BR, Peng ZZ, Li JS. Morphological characteristics and terminating patterns of masseteric neurons of the mesencephalic trigeminal nucleus in the rat: An intracellular horseradish peroxidase labeling study. J Comp Neurol 303: 286–299, 1991.

Shigenaga Y, Yoshida A, Mitsuhiro Y, Doe K, Suemune S. Morphology of single mesencephalic trigeminal neurons innervating periodontal ligament of the cat. Brain Res 448: 331–338, 1988.

Stanek E, Cheng S, Takatoh J, Han B-X, Wang F. Monosynaptic premotor circuit tracing reveals neural substrates for oro-motor coordination. eLife 3: e02511, 2014.

6. PLOS authors have the option to publish the peer review history of their article (what does this mean?). If published, this will include your full peer review and any attached files.

Reviewer #1: No

---

## [Author Response · Author response to Decision Letter 0]

6 Aug 2022

Response to Reviewers

Reviewer #1: The study by Chen et al., provides a morphological characterization showing the convergent projection from the mesencephalic trigeminal nucleus (Vme) and medial vestibular nucleus (MVN) onto motorneurons of the III and IV nuclei, as well as onto pre-motorneurons of the interstitial nucleus of Cajal (INC) and Darkschewitsch nucleus (DN).

The study in general is very well done. Combination of anterograde, retrograde and immnunostaining provides convincing experimental evidence supporting the conclusion that motorneurons and premotorneurons receive convergent innervation from the Vm and the MVN. This conclusion is supported by confocal microscopy showing presynaptic buttons from Vme and MVN neurons contacting the same cell body or dendrite, rising interesting perspectives on how these postsynaptic neurons integrate these inputs to control eye movements. Despite of that, some concerns arise.

Answer: we appreciate the reviewer’s valuable comment.

Major concerns:

Reviewer: Anterograde tracer (BDA) injected in the Vme yields a strong staining of the ipsilateral III and IV motor neuclei, as well as of the INC and DN. However, these injections yield only very weak staining of the trigeminal motor nucleus (Vmot), as shown in Figures 1F, 2A and 5D. This is surprising as Vme afferents (particularly spindle afferents) heavily project to the ipsilateral Vmot. This monosynaptic projection is very well documented in rodents, both electrophysiologically (Grimwood et al. 1992) and morphologically (Dessem et al. 1997; Luo and Li 1991; Luo et al. 1991; Stanek et al. 2014). Authors should provide an interpretation of this unexpected results of their experiments.

Answer: yes, the reviewer is correct, a large portion of Vme neuronal projections are going to Vmot. However, the projections from the Vme to the Vmot are actually collaterals of Vme neuronal central processes, based on results from intracellular horseradish peroxidase (HRP) labeling of masseter afferents by Dr. Shigenaga and colleagues [Shigenaga et al. 1988a, 1990]. The pseudo polar of single Vme neurons bifurcate to central and peripheral processes sooner or later away from the soma, the central axons consequently divide into a numerous collateral traveling to the Vmot, the supratrigeminal nucleus, the principle trigeminal sensory nucleus, the oralis of spinal trigeminal nucleus and the interstitial trigeminal regions and so on. Hence, the tract of tracer had been split into many different directions with a part of them reached the Vmot; so, the actual amount of tracer eventually arrived the Vmot may not be as much as we have expected.

On the other hand, the high molecular weight BDA travel slow through axons and degrade slow [Reiner et al. 2000]. We didn’t find published data about its degradation speed, but we believe it would degrade whenever they reached the destination, soma or terminal, like all other kinds of tracers those won’t stay there forever. The BDA would arrive the soma or bouton in the Vmot much earlier and degrade earlier. The distance between the Vme and Vmot is almost 1/6-8 of the distance between the caudal Vme and III based on a rat brain atlas [Paxinos and Weston 1998]. Thus, when the BDA densely filled in boutons at the III/IV and INC/DN, the tracer in the Vmot have been significantly degraded. 

Aforementioned 2 reasons may help to interpret the phenomenon that the BDA labeling in the Vmot is much weaker comparing to that in the III/IV and INC/DN. 

Reviewer: Results are discussed assuming that stained Vme neurons are only spindle afferents. However, the Vme also contain a large population of afferents innervating the periodontal ligament (Shigenaga et al. 1988). Moreover, the caudal portion of the Vme, where BDA injections were performed in this study by Chen and co-workers, contain a large proportion of these periodontal afferents. In that sense, most probably some of the anterogradely stained Vme afferents belong to this class. This possibility should be taken into consideration in the Discussion section.

Answer: right, the reviewer is correct. At the beginning, we discussed possible involvement of the periodontal afferent Vme neurons in this pathway. We cited a clinic entity of eyeball elevation by brushing teeth or squeezing gums [Gottlob et al., 2002], observed in a case of Marcus Gunn syndrome --- a congenital ptosis [Pratt et al., 1984; Brodsky CM, 2010]. However, we finally removed this part of discussion, as we were afraid that the readers would be confused, since our current study was not related to any neuro-ophthalmologic problem. While, as reviewer has indicated, the fact that the Vme, especially the caudal Vme, containing a large population of periodontal ligament afferent neurons [Shigenaga et al., 1988b; Nomura and Mizuno, 1985] could not be avoid. Therefore, we added a statement in Discussion (line 257- 260) as the follow: “However, the periodontal afferent Vme neurons could not be excluded since they are predominantly located in caudal part of the Vme [41], the area we injected the tracer. While, we considered jaw muscle afferent Vme neurons herein only because we had evidences that those neurons are involved in the related pathway [25-27].” 

Reviewer: Minor concerns:

For consistency capitalize letters identifying panels in Figure 1.

Line 66: citation “2” should be between square brackets.

Lines 72-73: citations “16-19” should be between square brackets.

Line 120: citation “35” should be between square brackets.

Line 222: citation “35” should be between square brackets.

Line 249: should read “nucleus of posterior commissure”

Line 293: citation “46” should be between square brackets.

Answer: thank the reviewer very much for her/his time. Following the reviewer’s guide, we found some problems in our EndNote, so we have made efforts to fix the problem. This time we have carefully examined each line to avoid the same kind of mis-format.

Correspondingly, we have capitalized the letters identifying the panels in the Fig. 1. 

References: 

Shigenaga Y et al., Morphology of single mesencephalic trigeminal neurons innervating masseter muscle of the cat.

Brain Res 1988a, 445:392-399.

Shigenaga Y et al., Two types of jaw-muscle spindle afferents in the cat as demonstrated by intra-axonal staining with HRP. Brain Res 1990, 514:219-237.

Reiner A et al., Pathway tracing using biotinylated dextran amines. J Neurosci Method 2000, 103:23-37.

Paxinos G and Weston C, The rat brain in stereotaxic coordinates. 1998, Springer, London, Boston, New York, Sydney, Tokyo, Toronto. 

Gottlob I et al., Elevation of one eye during tooth brushing. Am J Ophthalmol 2002, 134:459-460.

Pratt SG et al., The Marcus Gunn phenomenon. A review of 71 cases. Ophthalmology 1984, 91:27-30.

Brodsky CM, Pediatric Neuro-Ophthalmology. Second ed. New York, Dordrecht, Heidelberg, London: Springer, 2010.

Shigenaga Y, Yoshida A, Mitsuhiro Y, Doe K, Suemune S. Morphology of single mesencephalic trigeminal neurons innervating periodontal ligament of the cat. Brain Res 1988b, 448: 331–338.

Nomura S and Mizuno N, Differential distribution of cell bodies and central axons of mesencephalic trigeminal nucleus neurons supplying the jaw-closing muscles and periodontal tissue: a transganglionic tracer study in the cat. Brain Res 1985, 359:311-319.

---

## [Decision Letter · Decision Letter 1]

7 Sep 2022

PONE-D-21-39088R1Convergent innervations of mesencephalic trigeminal and vestibular nuclei neurons onto oculomotor and pre-oculomotor neurons --- tract tracing and triple labeling in ratsPLOS ONE

Dear Dr. Zhang,

Thank you for submitting your manuscript to PLOS ONE. After careful consideration, we feel that it has merit but does not fully meet PLOS ONE’s publication criteria as it currently stands. Therefore, we invite you to submit a revised version of the manuscript that addresses the points raised during the review process.

Please, find attached the comments of an expert second reviewer. As you will see, while some comments are simply issues of writing, others are important methodological clarifications that need to be addressed.A final decision concerning the acceptance of your manuscript will be made after the receipt of your revision and my evaluation of your responses to the reviewer's comments and critiques.

We look forward to receiving your revised manuscript.

Kind regards,

Sebastian Curti

Guest Editor

PLOS ONE

Journal Requirements:

Reviewers' comments:

Reviewer's Responses to Questions

**Comments to the Author**

1. If the authors have adequately addressed your comments raised in a previous round of review and you feel that this manuscript is now acceptable for publication, you may indicate that here to bypass the “Comments to the Author” section, enter your conflict of interest statement in the “Confidential to Editor” section, and submit your "Accept" recommendation.

Reviewer #2: (No Response)

2. Is the manuscript technically sound, and do the data support the conclusions?

Reviewer #2: Yes

3. Has the statistical analysis been performed appropriately and rigorously? 

Reviewer #2: Yes

4. Have the authors made all data underlying the findings in their manuscript fully available?

Reviewer #2: Yes

5. Is the manuscript presented in an intelligible fashion and written in standard English?

Reviewer #2: Yes

6. Review Comments to the Author

Reviewer #2: General concerns:

The general aim of the work of Chen et al is to explore the morphological bases of the circuit of the vestibular-ocular reflex in male and female rats by using different anterograde and retrograde tracers ionophoretically microinjected in the mesencephalic trigeminal and oculomotor nuclei. The experimental exploration of the hypothesis is important to the field of gravity-related research.

I think that the manuscript would be acceptable for publication if several important details in the Material and methods section were included in order to clarify the experimental procedures.

Specific concerns:

Abstract

Page 2:

Line 28: the correct form would be “vestibular-ocular reflex”; please put a synonymous of clearer: evident, easy to understand, etc.

Line 29: gravity-related (the same in page 4, line 97)

Line 38: please include the name of the two anterograde tracers and that motoneurons were immunostained with ChAT.

Line 43: We hypothesize that “the jaw muscle proprioceptive Vme…”

Introduction

Page 3-5

Line 60: please put a synonymous of “more demonstrated”, like established, elucidated, etc.

Line 89: please change “it is not impossible” for “a possibility is that…”

Line 91: please change “displayed” for “showed”

Line 104: please include the information about the immunostaining of the motoneurons with ChAT.

Materials and methods

-Please include the exact number of male and female rats used in all the study and in each of the different experimental groups. It is not clear established in this section that you use the same animals for BDA and PHAL injections.

-Please explain that the same animals receive the three microinjections of the tracers and if you did them consecutively or not

-Please explain if the same group of rats that receive the tracer microinjection, were used for the immunostaining procedure in order to put in evidence cholinergic motoneurons.

-Please explain if a third group of rats receive BDA and PHAL and then Ctb. Include the numbers and sex of the animals.

-I think that would be better if you refer to the different nuclei as right and left nucleus in order to clarify the methods, instead of contralateral or opposite. Such terms became confuse to the reader.

-An important point is related to the coordinates that you used to administered the tracers: why did you use a range instead of an exact localization of the site of each nuclei? i.e: line 120: 0.4-1.0 mm, 1.4-1.6 mm??

- Please describe how you determine that the microinjections were successfully and also, describe the percentage of success in the total number of animals.

Page 5

Line 110: please put the number of male and females in ordinal numbers (not 2/3, 1/3).

Line 116: Please include: did you use male, females or both? Number of each of them?

Line 118: please explain better the microinjection procedure and if the glass micropipette was connected to an iontophoretic pump.

Line 120: please include the reference of Paxinos and Watson (1998) atlas. Usually, stereotaxic coordinates in this type of studies were included as AP (anterior-posterior), DV (dorso-ventral) and L (lateral).

Lines 121 and 127: Please correct to the exact form: 16 degrees or 16º.

Line 123: only one time the animals receive analgesic?

Line 126: “contralateral MVN”. I suppose that it is with respect of Vme microinjection?

Page 6

Line 134: the frozen sections were collected in what solution? Did you store them at 4º or -20ºC?

Line 136: correct 8ºC

Lines 136 to 139: please explain better about the use of Alexa fluor 586-conjugated streptoavidin to reveal BDA, and so on.

Line 138: about 2 hours or 2 hours exactly?

Page 7

Line 166-167: please put in extensive the name of the ingredients/substances of the “cocktail”. Please note that such ingredients were not all secondary antibodies.

Results

Page 7

Line 185: distributed

Line 186: the representative case was male or female?

Did you evaluated the results obtained in both sex in a separate form?

Page 8:

Line 187: please correct this part: “without constant distributive preferential”

Line 213: we performed…

Line 214: in coronal sections obtained from rats that receive BDA and PHAL tracers.

Figure 1:

Legend: Distribution of Vme…. In the III/IV nucleus in a representative male/female animal.

Please use a, b, c or A, B, C…not both.

Please include the coordinates when you refer to the top or middle planes.

The labeled endings prefer to situate?

Please include a calibration bar to the microphotographs.

Figure 2:

Please include: in a representative male/female animal.

Microphotograph A is the same as F (figure 2)?

It would be better to understand if you include all the microphotographs with the dorsal part of the section at the top and the ventral one at the botton (like C or D)

Figure 3:

It would be better: Convergent innervations from Vme and MVN onto ChAT immunostained III/IV motoneuron

Figure 6:

Please include a smaller diagram.

I think that would be more appropriate if you perform a rat diagram, based in the results that you obtained. It is very speculative to expose that such mechanism and circuits would function in humans.

Please explain your working hypothesis in the figure legend based in this diagram. It mus be self-explicative.

7. PLOS authors have the option to publish the peer review history of their article (what does this mean?). If published, this will include your full peer review and any attached files.

Reviewer #2: **Yes: **Patricia Lagos

---

## [Author Response · Author response to Decision Letter 1]

15 Sep 2022

Reviewer #2: General concerns:

The general aim of the work of Chen et al is to explore the morphological bases of the circuit of the vestibular-ocular reflex in male and female rats by using different anterograde and retrograde tracers ionophoretically microinjected in the mesencephalic trigeminal and oculomotor nuclei. The experimental exploration of the hypothesis is important to the field of gravity-related research.

I think that the manuscript would be acceptable for publication if several important details in the Material and methods section were included in order to clarify the experimental procedures.

Response: Thank you for your valuable comment. 

Specific concerns:

Abstract

Page 2:

Line 28: the correct form would be “vestibular-ocular reflex”; please put a synonymous of clearer: evident, easy to understand, etc.

Response: Thank you. We have changed it to “vestibulo-ocular reflex”, in consistent to all previous publications (please see reference No. 8, 12, 14 and 15). 

Line 29: gravity-related (the same in page 4, line 97)

Response: Thank you. We have changed that in line 29, line 66, line 97, line 257 and line 339 of new-submitted manuscript.

Line 38: please include the name of the two anterograde tracers and that motoneurons were immunostained with ChAT. 

Response: I am afraid adding these words will increase word-count too much in Abstract, because the name of tracers is very long. Meanwhile, adding the vocabulary of tracers and antibody doesn’t change the meaning so much. 

Line 43: We hypothesize that “the jaw muscle proprioceptive Vme…”

Response: The sentence has been changed as the follow: ”We consider that jaw muscle proprioceptive Vme neurons projecting to the III/IV and INC ··· ··· ···”, because we had some evidences in our previous studies those can’t be cited herein. Our major hypothesis is that projections from masticatory afferent Vme neurons to the oculomotor and pre-oculomotor neurons may be a new source of “somatosensory vestibular interaction” related to the vertical-torsional VOR. 

Introduction

Page 3-5

Line 60: please put a synonymous of “more demonstrated”, like established, elucidated, etc.

Response: The word “elucidated” is better, we changed it to “more elucidated”. Thank you.

Line 89: please change “it is not impossible” for “a possibility is that…”

Response: We have changed it to “it is possible that”.

Line 91: please change “displayed” for “showed”

Response: Changed, thank you. 

Line 104: please include the information about the immunostaining of the motoneurons with ChAT. 

Response: If the terminology of motoneuron marker is added here, the way and name of marker to identify the pre-oculomotor neuron should be added in parallel, which won’t change the key meaning of the paragraph but make the context redundant. 

Materials and methods

-Please include the exact number of male and female rats used in all the study and in each of the different experimental groups. It is not clear established in this section that you use the same animals for BDA and PHAL injections.

Response: We have added a Table 1 in which we described experimental group division, and introduced what tracer and where the tracer is injected. Please see Table 1 for detail.

-Please explain that the same animals receive the three microinjections of the tracers and if you did them consecutively or not

Response: Yeah, please see Table 1 for detail.

-Please explain if the same group of rats that receive the tracer microinjection, were used for the immunostaining procedure in order to put in evidence cholinergic motoneurons.

Response: Yeah, please see Table 1 for detail.

-Please explain if a third group of rats receive BDA and PHAL and then Ctb. Include the numbers and sex of the animals.

Response: Yes, you are right. In the third group we injected BDA, PHA-L and CTB in the same rats. We have described the procedure of triple labeling in page 7 line 158 – 161. In addition, we have further clarified it in the new Table 1, so please see Table 1 for detail.

-I think that would be better if you refer to the different nuclei as right and left nucleus in order to clarify the methods, instead of contralateral or opposite. Such terms became confuse to the reader.

Response: Yeah, sorry for the confuse. We did perform injection in either right or left side, as we used either male or female animals, in preliminary test. When it is clarified that there is no difference in side and sex, we’ll keep injecting one side. However, using “unilateral”, “ipsilateral” and “contralateral” is a common expression to describe different injection site, lesioned site and/or the resultant labeled area in the same animal in neuroanatomic studies. 

-An important point is related to the coordinates that you used to administered the tracers: why did you use a range instead of an exact localization of the site of each nuclei? i.e: line 120: 0.4-1.0 mm, 1.4-1.6 mm??

Response: Good question. Even in the same litter animals, some grow fast and some develop slowly; so, their brain and body sizes are different. While, the brain and body sizes are more variated in different litter of rats. Hence, the stereotaxic coordinates have to be adjusted a little bit based on body weight and distance between the bregma and interaural line. 

- Please describe how you determine that the microinjections were successfully and also, describe the percentage of success in the total number of animals.

Response: Originally, we have delineated our criteria to judge a good injection targeted to a certain nucleus, such as “A vigorous lower jaw elevation by minimum iontophoresis current is a sign of correct injection” for a good injection to the Vme, and so on. After revision, we moved those descriptions into the Table 1, to make it more noticeable. 

Page 5

Line 110: please put the number of male and females in ordinal numbers (not 2/3, 1/3).

Response: Yes, we have changed as you suggested.

Line 116: Please include: did you use male, females or both? Number of each of them?

Response: Yeah, please see line 110 and Table 1.

Line 118: please explain better the microinjection procedure and if the glass micropipette was connected to an iontophoretic pump.

Response: We have rephrased the relative part like this: “The BDA was iontophoretically delivered with 2 Hz, 200 ms duration positive current output by an electric stimulator (Grass Pulse Stimulator S88, A-M Systems, Sequim, WA).” Please read line 122 - 124 in new-submitted revised-manuscript.

Line 120: please include the reference of Paxinos and Watson (1998) atlas. Usually, stereotaxic coordinates in this type of studies were included as AP (anterior-posterior), DV (dorso-ventral) and L (lateral).

Response: Please read the manuscript from line 121 to 122, from line 128 to 130 and from line 161 to 163, the “No. 35” reference is the atlas edited by Paxinos and Watson. And “XX mm posterior (or anterior) to the interaural line”, “XX mm posterior (or anterior) to the bregma” and “XX mm lateral to the midline” are all based on the coordinates shown in the atlas. So, what should we revise? 

Lines 121 and 127: Please correct to the exact form: 16 degrees or 16º.

Response: What is the difference? What should we change? 

Line 123: only one time the animals receive analgesic? 

Response: Thank you for reminding, we have rephrased it to “The animals were administered analgesic after surgery for 3 days”, showing in line 125 in new-submitted manuscript.

Line 126: “contralateral MVN”. I suppose that it is with respect of Vme microinjection?

Response: Correct, we have rephrased the sentence like this: “PHA-L was delivered to the MVN contralateral to the BDA injection to the Vme, with a coordinate that ··· ··· ···”. That is in line 128 – 129 of new-submitted manuscript.

Page 6

Line 134: the frozen sections were collected in what solution? Did you store them at 4º or -20ºC?

Response: Cutting and colleting sections is a very routine procedure that is known to almost all authors. Anyway, we have rephrased sentence like this “Frozen sections were cut into 30 �m and collected serially from caudal to rostral in 0.01 M PBS (pH 7.2-7.4) in room temperature (RT). See new-submitted manuscript line 135- 136. 

Line 136: correct 8ºC

Response: It shows as 8oC in my computer, is it a problem? 

Lines 136 to 139: please explain better about the use of Alexa fluor 586-conjugated streptoavidin to reveal BDA, and so on.

Response: This is a routine procedure for immuno-fluorescent visualization of BDA. We are not sure what we should revise here? We apologize for our misunderstanding, if we did. 

Line 138: about 2 hours or 2 hours exactly?

Response: Yeah, about 2 hours. The sentence has been rephrased as “the sections were transferred into a cocktail ∙ ··· ··· ···, and incubated for about 2 hours to visualize triple labeling”. See new-submitted manuscript line 138 - 141. 

Page 7

Line 166-167: please put in extensive the name of the ingredients/substances of the “cocktail”. Please note that such ingredients were not all secondary antibodies.

Response: We have rephrased the sentence as the follow: “the sections were incubated in the cocktail of the same conjugated 594 and 488, plus donkey anti-goat conjugated either Alexa Fluor 350 or ByLight 405 to reveal CTB labeled pre-oculomotor neurons; however, the Alexa Fluor 350 was for conventional fluorescent microscopy and the ByLight 405 was for confocal microscopy observations”. See new-submitted manuscript line 167-170.

Results

Page 7

Line 185: distributed

Response: Corrected, in new-submitted line 187, thank you.

Line 186: the representative case was male or female?

Did you evaluated the results obtained in both sex in a separate form?

Response: We are sorry we have forgotten the sex of this animal; Drs Chen and Liang are going to find old notebook. However, we think it doesn’t matter for the sex of the animals, since the distribution of the anterograde labeling looked no obvious difference between female and male. Generally, neural tract tracing study is a kind of qualitative study, rather than quantitative measurement.

Page 8:

Line 187: please correct this part: “without constant distributive preferential”

Response: Thank you. We have rephrased this sentence as “The labeled fibers and boutons in the IV appear to be evenly distributed in the coronal planes”. See line 193-194 of new-submitted manuscript. 

Line 213: we performed…

Response: Yeah, it has been revised (line 215 in new-submitted manuscript).

Line 214: in coronal sections obtained from rats that receive BDA and PHAL tracers.

Response: We are not sure what should be changed here. Anyway, thank you for your time.

Figure 1:

Legend: Distribution of Vme…. In the III/IV nucleus in a representative male/female animal.

Response: We are sorry, we forgot this figure is from a male or female rat. But by our observations, there was no sexual difference for the labeling as aforementioned.

Please use a, b, c or A, B, C…not both.

Response: Yeah, thank you. We have replaced “a -- f” with “A - F” from line 189 to 195 at page 8 of new-submitted manuscript. 

Please include the coordinates when you refer to the top or middle planes.

Response: People commonly use the coordinates in live animals. It is not easy to match brain sections to stereotaxic coordinates! Even though it is possible, what is the pursers for doing so? 

The labeled endings prefer to situate?

Response: We have rephrased the sentence as the follow: “the labeled endings are preferentially distributed in ventral and ventromedial part of the III in the coronal sections”. Thank you.

Please include a calibration bar to the microphotographs.

Response: We are sorry, we didn’t purchase software with reconstruction and measurement or calibration function. We just made some screen-shot when Camera Lucida tube transfer the imaging to a computer screen. The purpose of figure 1 is to show a distributive patter of BDA labeled axons and terminals. 

Figure 2:

Please include: in a representative male/female animal.

Response: This rat was from a female rat. As I remember the last rat in this group is a female, and the photos were from the last one in this group. Even though, we had better not give a sex information because we can’t remember all of them unless Dr Liang, who has already retired, could find old notebook. 

Microphotograph A is the same as F (figure 2)?

Response: We guess your question is whether or not these pictures were taken from the same animal, right? Yes, if it is the question. 

It would be better to understand if you include all the microphotographs with the dorsal part of the section at the top and the ventral one at the botton (like C or D)

Response: Yes, you are right. That is common regulation in almost all animal’s brain atlas, and in all neuroanatomical papers. So, nothing special in Fig. 2 C and D. 

Figure 3:

It would be better: Convergent innervations from Vme and MVN onto ChAT immunostained III/IV motoneuron

Response: We have rephrased it as you suggested. Thank you. 

Figure 6:

Please include a smaller diagram.

I think that would be more appropriate if you perform a rat diagram, based in the results that you obtained. It is very speculative to expose that such mechanism and circuits would function in humans.

Please explain your working hypothesis in the figure legend based in this diagram. It mus be self-explicative.

Response: There exists a large extent of similarity between rat’s neuronal circuit and human brain networks, which is reflected by that rats were and are broadly used in neuroscience studies to mimic human’s brain structures and pathology, and is substantiated by some comparative investigations [Ma Z et al. Neuroimage 2018, 170:95-112; Toossi et al. Sci Rep 2021, 11:1955]. Report on comparison of vestibulo-ocular system between rat and monkey is few [Schuerger and Balaban: Brain Res Rev 1999, 30:189-217], but similar neuronal connections was also observed. Likewise, we drew a schematic of human head with partial masticatory and oculomotor system, attempt to use our finding in rat to explain certain potential neurological mechanism in human. But, it doesn’t mean rat has exactly the same head structure and posture as human. Humans are bipedal erect animals with face ahead and the face top to the neck; however, rats are quadrupedal animal with face lateral and the face anterior to the neck. Therefore, head posture in rats and humans is largely different, drawing a schematic in rat head won’t help reader to understand the key topic; in contrary, it would distract readers from understanding the core topic.

---

## [Editor Report · Decision Letter 2]

17 Oct 2022

PONE-D-21-39088R2Convergent innervations of mesencephalic trigeminal and vestibular nuclei neurons onto oculomotor and pre-oculomotor neurons --- tract tracing and triple labeling in ratsPLOS ONE

Dear Dr. Zhang,

Thank you for submitting your manuscript to PLOS ONE. After careful consideration, we feel that it has merit but does not fully meet PLOS ONE’s publication criteria as it currently stands. Therefore, we invite you to submit a revised version of the manuscript that addresses the points raised during the review process.

**Required changes:**

While I agree with the idea of a table in order to summarize data, the way Table 1 is constructed is confusing. It would be much valuable a summary of the different experiments with its corresponding successful rates, disregarding of the sex of the animals If this wasn’t systematically annotated. Include the number of experiment and the number of successful observations, for instance in how many experiments injecting BDA at Vme+PHA-L at MVN+Nissl did authors observed convergent innervation of III/IV neurons. Below is a posible organization for the table.

Experiment

Observation

Total number of experiments

BDA@Vme

axon terminals III/IV nuclei (n=x)

BDA@Vme+PHA-L@MVN+Nissl

convergent innervation of III/IV neurons (n=x)

BDA@Vme+PHA-L@MVN+ChAT immune

convergent innervation of III/IV motoneurons

BDA@Vme+PHA-L@MVN 

INC

BDA@Vme+PHA-L@MVN+CTB@III

convergent innervation of INC neurons

The criterion employed to consider a given injection as a successful one should be included in the text of the Materials and Methods section.

According to the author’s response to the second Reviewer, in a preliminary set of experiments the authors determined that injections in either right or left side, as well as in male or female animals, yield no difference and hence injections were done in either side and in animals of either sex. If this is the case, this should be stated in the Materials and Methods section.

The Figure 6 and associated discussion should be reformulated according to comments of the second Reviewer. While I agree that exists a large extent of similarity between rat’s neuronal circuit and human brain networks, the great difference of head posture between bipeds and quadrupeds, as recognized by the authors themselves, makes most of the conclusions highly speculative. Accordingly, in the Discussion section, extrapolation of findings in rat to explain neurological mechanism in human should be carefully done, and must include explicit considerations and limitations due to postural difference as mentioned above.

**Recommended changes: **

Line 68 (resubmitted manuscript): The expression *“VOR reflexive arc”* is redundant. Also, clarify the whole sentence. A possible rephrasing: *“Studies of VOR and its underlying mechanisms revealed that neuronal circuits executing horizontal…”*Line 75: Change to “participate in the perception of” or “sense”, “perceive” at it is stated imply a higher cognitive process carried by the individual.Line 100: “These facts infer that the pathway is possibly…” Please rephrase “suggest” or “indicate” instead of “infer”.Line 106: “Ishii, the author, concluded that alternative change…” This expression is colloquial, please rephrase.“…,in addition to the VN efferent innervations onto the masseter motoneurons [21].” I think the use of plural is incorrect, check please.Line 111: “reaching the oculomotor system may be interacted by aforementioned…” not clear, perhaps “integrated with” instead.Line 146: “, in the same parameter as aforementioned,” This refers to the ionophoretic injection parameters? If so, state it explicitly please.Line 200: The previous conjugated is 568 and not 594, please check. 

We look forward to receiving your revised manuscript.

Kind regards,

Sebastian Curti

Guest Editor

PLOS ONE
---

## [Author Response · Author response to Decision Letter 2]

7 Nov 2022

Reviewer’s commend and our responses:

1. While I agree with the idea of a table in order to summarize data, the way Table 1 is constructed is confusing. It would be much valuable a summary of the different experiments with its corresponding successful rates, disregarding of the sex of the animals If this wasn’t systematically annotated. Include the number of experiment and the number of successful observations, for instance in how many experiments injecting BDA at Vme+PHA-L at MVN+Nissl did authors observed convergent innervation of III/IV neurons. Below is a posible organization for the table.

Experiment Observation Total number of experiments

BDA@Vme axon terminals III/IV nuclei (n=x) 

BDA@Vme+PHA-L@MVN+Nissl convergent innervation of III/IV neurons (n=x) 

BDA@Vme+PHA-L@MVN+ChAT immune convergent innervation of III/IV motoneurons 

BDA@Vme+PHA-L@MVN INC 

BDA@Vme+PHA-L@MVN+CTB@III convergent innervation of INC neurons 

Response: We have revised the Table 1 based on the above format, and in the new Table 1, the success rate was listed since the previous review required to “describe the percentage of success in the total number of animals”. Please see new Table 1. 

2. The criterion employed to consider a given injection as a successful one should be included in the text of the Materials and Methods section.

Response: Yeah, we have rephrased relative section of the Materials and Methods as you suggested. Please see the new-submitted manuscript in page 5, line 124 - 125 and line 131 - 132; and in page 7 line 165 - 166. 

3. According to the author’s response to the second Reviewer, in a preliminary set of experiments the authors determined that injections in either right or left side, as well as in male or female animals, yield no difference and hence injections were done in either side and in animals of either sex. If this is the case, this should be stated in the Materials and Methods section.

Response: Yeah, we have added a new section in the Materials and Methods named as “Sexual and side difference in BDA, PHA-L and CTB injections”. Please see the new-submitted manuscript in page 7, from line 174 to 177. 

4. The Figure 6 and associated discussion should be reformulated according to comments of the second Reviewer. While I agree that exists a large extent of similarity between rat’s neuronal circuit and human brain networks, the great difference of head posture between bipeds and quadrupeds, as recognized by the authors themselves, makes most of the conclusions highly speculative. Accordingly, in the Discussion section, extrapolation of findings in rat to explain neurological mechanism in human should be carefully done, and must include explicit considerations and limitations due to postural difference as mentioned above.

Response: We have added “a smaller rat diagram” in the Fig. 6 as suggested by the last reviewer. Correspondently, we diagrammatized location and extension direction of the rat’s jaw and neck muscles, and described the directions of different potential vectors during head rolling. Please see Fig. 6A in new-submitted manuscript, and the relevant discussion in page 12 line 285-286 and line 302-303. 

Recommended changes: 

1. Line 68 (resubmitted manuscript): The expression “VOR reflexive arc” is redundant. Also, clarify the whole sentence. A possible rephrasing: “Studies of VOR and its underlying mechanisms revealed that neuronal circuits executing horizontal…”

Response: Thank you for the suggestion. That has been amended (see page 3 line 58). 

2. Line 75: Change to “participate in the perception of” or “sense”, “perceive” at it is stated imply a higher cognitive process carried by the individual.

Response: Yeah, we have changed this part. Please see the new-submitted manuscript page 3 line 65-66. 

3. Line 100: “These facts infer that the pathway is possibly…” Please rephrase “suggest” or “indicate” instead of “infer”.

Response: Thanks. We have replaced “infer” with “suggest”. Please see the new-submitted manuscript page 4 line 87. 

4. Line 106: “Ishii, the author, concluded that alternative change…” This expression is colloquial, please rephrase.

Response: Yeah, the sentence has been changed as the follow: “The author concluded that ∙∙∙ ∙∙∙ ∙∙∙” (Line 93). 

5. “…,in addition to the VN efferent innervations onto the masseter motoneurons [21].” I think the use of plural is incorrect, check please.

Response: Thank you, amended (see new-submitted page 4 line 96). 

6. Line 111: “reaching the oculomotor system may be interacted by aforementioned…” not clear, perhaps “integrated with” instead.

Response: Thank you for this direction, we have rephrased this sentence as you suggested. Please read new-submitted manuscript from line 97 to 100 in page 4. 

7. Line 146: “, in the same parameter as aforementioned,” This refers to the ionophoretic injection parameters? If so, state it explicitly please.

Response: Yeah, this part was also rephrased as you suggested as the following: “2.5% PHA-L ··· ··· ···, was delivered with 2 Hz, 200 ms duration positive current output to the MVN ··· ··· ···” (see page 5, line128 – 129 of new-submitted). 

8. Line 200: The previous conjugated is 568 and not 594, please check.

Response: Thank you and it was amended (see page 7, line 171 of new-submitted).

---

## [Editor Report · Decision Letter 3]

14 Nov 2022

Convergent innervations of mesencephalic trigeminal and vestibular nuclei neurons onto oculomotor and pre-oculomotor neurons --- tract tracing and triple labeling in rats

PONE-D-21-39088R3

Dear Dr. Zhang,

We’re pleased to inform you that your manuscript has been judged scientifically suitable for publication and will be formally accepted for publication once it meets all outstanding technical requirements.

Kind regards,

Sebastian Curti

Guest Editor

PLOS ONE
---

## [Editor Report · Acceptance letter]

16 Nov 2022

PONE-D-21-39088R3 

Convergent innervations of mesencephalic trigeminal and vestibular nuclei neurons onto oculomotor and pre-oculomotor neurons --- tract tracing and triple labeling in rats 

Dear Dr. Zhang:

I'm pleased to inform you that your manuscript has been deemed suitable for publication in PLOS ONE. Congratulations! Your manuscript is now with our production department. 

Kind regards, 

on behalf of

Dr. Sebastian Curti 

Guest Editor

PLOS ONE